# A Study on the Effect of Various Media and the Supplementation of Organic Compounds on the Enhanced Production of Astaxanthin from *Haematococcus lacustris* (Girod—Chantrans) Rostafinski (Chlorophyta)

**DOI:** 10.3390/microorganisms12061040

**Published:** 2024-05-21

**Authors:** Vijay Rayamajhi, Yunji An, Huijeong Byeon, Jihyun Lee, Taesoo Kim, AhJung Choi, JongDae Lee, KwangSoo Lee, ChulHyun Kim, HyunWoung Shin, SangMok Jung

**Affiliations:** 1Department of Biology, Soonchunhyang University, Asan 31538, Chungcheongnam-do, Republic of Korea; 2Korea Fisheries Resources Agency East Sea Branch, Samho-ro, Buk-gu, Pohang 37601, Gyungsangbuk-do, Republic of Korea; 3Department of Environmental Health Science, Soonchunhyang University, Asan 31538, Chungcheongnam-do, Republic of Korea; 4Department of Sports Science, Soonchunhyang University, Asan 31538, Chungcheongnam-do, Republic of Korea; 5Department of Sports Medicine, Soonchunhyang University, Asan 31538, Chungcheongnam-do, Republic of Korea; 6AlgaeBio, Inc., Asan 31459, Chungcheongnam-do, Republic of Korea; 7Research Institute for Basic Science, Soonchunhyang University, Asan 31538, Chungcheongnam-do, Republic of Korea

**Keywords:** microalgae, astaxanthin, leucine, sodium glutamate, *Haematococcus lacustris*, Jaworski’s medium

## Abstract

Natural astaxanthin is in high demand due to its multiple health benefits. The microalga *Haematococcus lacustris* has been used for the commercial production of astaxanthin. In this study, we investigated the effects of six different media with and without a nitrogen source and supplementation with nine organic compounds on the growth and astaxanthin accumulation of *H. lacustris*. The highest astaxanthin contents were observed in cultures of *H. lacustris* in Jaworski’s medium (JM), with a level of 9.099 mg/L in JM with a nitrogen source supplemented with leucine (0.65 g/L) and of 20.484 mg/L in JM without a nitrogen source supplemented with sodium glutamate (0.325 g/L). Six of the nine organic compounds examined (leucine, lysine, alanine, sodium glutamate, glutamine, and cellulose) enhanced the production of astaxanthin in *H. lacustris*, while malic acid, benzoic acid, and maltose showed no beneficial effects.

## 1. Introduction

Astaxanthin (3,3′-dihydroxy-β-carotene-4,4′-dione) is a red–orange pigment belonging to the carotenoid family with commercial applications in cosmetics, food supplements, livestock feedstuffs, and pharmaceuticals due to its remarkable biochemical characteristics, physiological effects, and physical properties [1,2]. Although astaxanthin is naturally produced by a range of fungal and bacterial species, *Haematococcus lacustris* (Girod-Chantrans) Rostafinski (Chlorophyta) is considered the chief producer of astaxanthin among microalgae with astaxanthin contents up to 5% of cell dry weight [3]. The large-scale production of astaxanthin from yeasts, fungi, bacteria, shrimp, fish, and so forth is not feasible because of their lower astaxanthin contents (less than 1% of dry weight) [4]. Among microalgae, edible astaxanthin is mainly produced by *Haematococcus lacustris*, *Halochlorella rubescens, Rhexinema sarcinoideum, Chromochloris zofingiensis*, and *Ettlia carotinosa* [5]. *Haematococcus lacustris* is an important microalga in terms of its astaxanthin production, biomass, and lipid production, which can be utilized for commercial purposes. Astaxanthin, a tetraterpene with linked isoprene units, consists of a linear polyene chain and two terminal β-ionone rings [1,6,7,8,9]. The presence of 13 conjugated double-bond systems causes it to have a pink and red color [1,6]. The global astaxanthin market value was estimated to be USD 1 billion in 2019, which is anticipated to reach USD 3398.8 million by 2027 [8]. Natural astaxanthin has properties that aid growth and enhance immunity as well as antioxidant, antiobesity, anticancer, antidiabetic, and antiinflammatory functions and protective effects against gastric ulcers and a number of ocular diseases. Due to these properties, it is used as a major bioactive component in beauty products, antiaging serums, and sunblock creams and is also used as an oral dietary supplement [6]. *H. lacustris* cells undergo a number of ultrastructural changes throughout their life cycle depending on exposure to various stress conditions. Typically, the cells are spherical–ovoid in shape, with a diameter of approximately 30 µm. Initially, they are green palmelloid or biflagellate, freely swimming with a single pyrenoid-containing chloroplast. Later, the cells lose their flagella, round up into a nonmotile palmella, and turn into thick-walled aplanospores. The green-stage vegetative cells can reproduce asexually and give rise to 2 to 32 daughter cells. The accumulation of astaxanthin starts during the intermediate stage between days 7 and 10 when *H. lacustris* cells turn greenish–orange. There is an increase in the cell size and the loss of flagella due to stress conditions. The “red-astaxanthin formation” (red stage) appears between 11 and 14 days, where astaxanthin accumulation continues, with the cells forming cysts at the aplanospore stage [10]. Then, a thick algeenan-containing cell wall is formed, which protects the aplanospore cells from acetolysis via high light exposure or nutrient deprivation. Finally, astaxanthin accumulates in droplets in the perinuclear cytoplasm, and the cells appear bright red in the mature aplanospore stage [10].

Some organic carbon sources, such as glucose and acetate, have been used for the supplementation of media in the commercial production of astaxanthin from *H. lacustris* (Girod-Chantrans) Rostafinski (Chlorophyta) [11]. Supplementation with sodium acetate as an organic carbon source increases *H. lacustris* growth and astaxanthin accumulation. The addition of organic carbon sources to phototrophic growth cultures influences photosynthesis and carbon metabolism in *H. lacustris* [12]. In a recent study, we evaluated the effects of four organic carbon sources: ribose, glycerol, sodium acetate, and sodium gluconate [5]. Astaxanthin accumulation is enhanced in *H. lacustris* by supplementation with glycerol [13], sodium fumarate [14], trisodium citrate [15], and oxaloacetate [16] under various conditions. Sodium glutamate, or monosodium glutamate (MSG), is a widely used food enhancer that was originally extracted from beet sugar but has also been obtained by the microbial fermentation of potato starch, cassava, wheat, and so forth [17]. Microalgae produce high levels of carotenoids when mariculture wastewater is modified with sodium glutamate residue [18]. Adding sodium glutamate to a medium is beneficial for the growth of *Spirulina maxima* (now classified as *Limnospira maxima*) [19,20]. Malic acid is a dicarboxylic acid with four carbons that has biological phytohormone effects and promotes cold resistance and the growth of seedlings [21]. Leucine is an important nutrient in plants that plays a role in signalling for lipid decomposition, protein metabolism, biomass accumulation, and other biological reactions [22]. Glutamine affects gene expression in plants, and at low concentrations, it has been shown to improve the growth of rice seedlings [23]. Benzoic acid inhibits plant growth and plays a major role in crop autotoxicity [24]. Cellulose is a simple polymer and the most abundant carbohydrate produced by plants. It is highly resistant to enzymatic hydrolysis and forms insoluble, crystalline microfibrils. A complete set of enzymes with the ability to effectively degrade cellulose is present only in a few microorganisms, such as eubacteria, fungi, anaerobic cellulose-degrading protozoa, avocado fruit, and slime mold (*Dictyostelium discoideum*) [25] Moreover, *Chlamydomonas reinhardtii*, a phototrophic unicellular green microalga belonging to the order Chlamydomonadales and the class Chlorophyceae, has the ability to utilize cellulose for its growth in the absence of other carbon sources and digest exogenous cellulose when grown under CO₂-limiting conditions in the light [26]. Alanine improves lipid production and biomass accumulation in microalgae [27]. Lysine plays a number of different metabolic roles in plants [28]. The effects of maltose have been studied in *Auxenochlorella pyrenoidosa* (formerly *Chlorella pyrenoidosa*), and it has been shown to increase the growth of *Botryococcus braunii* and the cell density of *A. pyrenoidosa* [29,30].

In this study, we used six different media that have been utilized for the growth of microalgae in recent studies. We compared and evaluated the effects of these media on the growth and astaxanthin accumulation of *Haematococcus lacustris* (Girod-Chantrans) Rostafinski (Chlorophyta) under laboratory conditions. As our preliminary experiments showed that *H. lacustris* produced more astaxanthin in Jaworski’s medium (JM) under nitrogen-depleted conditions compared to other media, this was used as the basal medium. The other consideration of using JM was its economical profitability. Next, we examined the effects of the supplementation of JM with different readily available organic compounds on growth and astaxanthin accumulation in *H. lacustris*.

## 2. Materials and Methods

### 2.1. Microalgae and Culture Conditions

*H. lacustris* (LIMS-PS-1354) was obtained from the Library of Marine Samples at The Korea Institute of Ocean Science and Technology (KIOST), Geoje, Republic of Korea, and cultured and maintained at 21 ± 1 °C under a 12/12 h light–dark cycle and a photon flux density of 37 µmol m⁻^2^ s⁻^1^ in (JM).

### 2.2. Preparation of Growth Media

Six different media were prepared with and without a nitrogen source for the experiment, as described in the cited references: JM [5], Walne/Conway [31], Optimal Haematococcus Medium (OHM) [32], Blue Green-11 (BG-11) [33,34,35,36], modified Chu-13 [37], and Bold’s Basal Medium (BBM) [38].

For the preparation of media without nitrogen sources, Walne/Conway, OHM, modified Chu-13, and JM were prepared without KNO_3_. Similarly, BG-11, BBM, and JM were prepared without NaNO_3_. The constituents of the six media are listed in Table 1. The pH of the modified CHU 13, BG-11, JM, BBM, Walne/Conway, and OHM were adjusted to 6.7, 7.5, 7.0, 6.8, 7.5, and 7.0, respectively. The experiments were carried out in three replicates in 40 mL vials containing 30 mL of the medium and 10 mL of the seed culture. A 4-day-old culture was used as the seed culture for the initial inoculum. The media experiment was replicated five times, and the experiments with the supplementation of organic carbon compounds were replicated three times. The experiment was carried out over a period of 14 days.

The effects of supplementation with the following nine organic compounds were examined: sodium glutamate, malic acid, leucine, glutamine, benzoic acid, cellulose, lysine, alanine, and maltose. All nine of these organic compounds are carbon sources, while five (alanine, leucine, lysine, glutamine, and sodium glutamate) are also nitrogen sources. These compounds were added to the media at concentrations of 0 g/L (control), 0.325 g/L, 0.65 g/L, 1.3 g/L, and 2.6 g/L.

### 2.3. Experimental Design

*H. lacustris* was cultured under a photon flux density of 40 µmol m^–2^ s^–1^. The fluorescent light intensity was measured using a light meter (Fluke-941; Fluke Corp., Everett, WA, USA). The experiments were performed at 16 °C ± 2 °C.

### 2.4. Analysis of the Cell Number and Astaxanthin Accumulation

A haemocytometer with an improved Neubauer chamber (Marienfeld Superior, Lauda-Konigshofen, Germany) and a light microscope with a 20× objective lens (BX53; Olympus, Tokyo, Japan) were used to estimate the cell number. The analysis of astaxanthin was performed according to the methods described in previous studies [5,6].

### 2.5. Chemicals and Reagents 

Benzoic acid, dl-malic acid, l-alanine, zinc chloride, ammonium molybdate tetrahydrate, chromium (III) oxide, cobalt (II) nitrate hexahydrate (Duksan Reagents, Asansi, Gyeonggi-do, Republic of Korea), cellulose, sodium carbonate, calcium chloride dihydrate, copper(II) sulphate pentahydrate, thiamine hydrochloride, biotin (Sigma-Aldrich, Steinheim, Germany), sodium l(+)-glutamate monohydrate, d(+)-maltose monohydrate, l(+)-glutamine, l-leucine, l-lysine monohydrochloride, potassium nitrate, sodium phosphate dibasic anhydrous, magnesium sulphate heptahydrate, iron (III) citrate hydrate, cyanocobalamin, thiamine, potassium phosphate dibasic anhydrous, chromium (III) oxide, calcium nitrate tetrahydrate, sodium phosphate tribasic anhydrous (Daejung Reagents Chemicals, Gyronhhi-do, Republic of Korea), ferric ammonium citrate (Kisanbio, Seoul, Seocho-gu, Republic of Korea), iron (III) chloride hexahydrate, cobalt (II) chloride hexahydrate (Kanto Chemical Industry Co., Ltd., Chiyoda-Ku, Tokyo, Japan), citric acid anhydrous (Samchun Pure Chemical Co., Ltd., Gangnam-gu, Seoul, Republic of Korea), and zinc sulphate heptahydrate (Junsei Chemical Co., Ltd., Chuo-ku, Tokyo, Japan) were obtained from the sources shown and used in the experiments as received.

### 2.6. Statistical Analysis 

The cell number and pigment data are represented as the mean ± standard deviation. Statistical analyses were performed using one-way ANOVA followed by Tukey’s multiple comparison test using SPSS (version 22; IBM Corp., Armonk, NY, USA). In all analyses, *p* < 0.001 was taken to indicate statistical significance.

## 3. Results

### 3.1. Effect of Media on the Growth of H. lacustris 

During the cultivation in BBM with a nitrogen source (+N), the *H. lacustris* cell cultures were green until day 9 but turned orange on day 12. However, the cultures remained green until day 12 when cultivated in BBM without a nitrogen source (–N) (Figure 1). During cultivation in JM (+N) and JM (−N), the cultures were green until day 6 but turned orange on day 9 and pink on day 12. During cultivation in BG-11 (+N), they were green until day 12. In BG-11 (−N), however, they remained green until day 9 but turned yellowish orange on day 12. During cultivation in Walne/Conway (+N), the cultures were green until day 9 and turned pink on day 12. In Walne/Conway (−N), they remained green until day 6, turned orange on day 9, and turned pink on day 12. In OHM (+N), they were green until day 12. In OHM (−N), they remained green until day 9 but turned yellowish–orange on day 12. In Chu-13 (+N), they were green until day 9 but turned yellowish–orange on day 12. In Chu-13 (−N), they remained green until day 9 and turned orange on day 12.

The initial cell numbers on day 0 for BBM (+N), BG-11 (+N), Chu-13 (+N), Conway (+N), JM (+N), and OHM (+N) were 5.48 × 10^4^ cells/mL, 2.69 × 10^4^ cells/mL, 3.63 × 10^4^ cells/mL, 4.4 × 10^4^ cells/mL, 3.09 × 10^4^ cells/mL, and 2.68 × 10^4^ cells/mL, respectively (Figure 2). On day 6, *H. lacustris* showed significant increases in the cell number in all media with a nitrogen source. Similarly, the initial cell numbers on day 0 for the culture without N were 2.47 × 10^4^, 2.47 × 10^4^, 2.57 × 10^4^, 2.47 × 10^4^, 2.01 × 10^4^, and 2.51 × 10^4^ cells/mL, respectively. The cell numbers increased significantly on days 6, 9, and 12 for all media. On day 12, there was an increase of 11.99-fold in the cell number for BBM (−N), 9.52-fold for BG-11 (−N), 11.69-fold for Chu-13 (−N), 10.12-fold for Conway (−N), 15.48-fold for JM (−N), and 7.80-fold for OHM (−N).

### 3.2. Effect of Media on Astaxanthin Levels in H. lacustris

The astaxanthin levels were the highest for JM (−N) and the lowest for OHM (+N). The astaxanthin production by *H. lacustris* grown in JM (−N) was approximately 15% higher than that in Conway/Walne (−N), 22% higher than that in Chu-13 (−N), 50% higher than that in BBM (−N), 183% higher than that in BG-11 (−N), and 158% higher than that in OHM (−N) (Figure 3).

### 3.3. Effects of Organic Compounds on H. lacustris Cell Numbers and Astaxanthin Levels

*H. lacustris* was grown in JM (+N) and JM (−N) supplemented with various organic compounds, and the cell numbers are shown in Table 2, Table 3, Table 4, Table 5, Table 6, Table 7, Table 8, Table 9, Table 10, Table 11, Table 12, Table 13, Table 14, Table 15, Table 16, Table 17, Table 18 and Table 19. When grown in JM (+N), the cell numbers increased significantly under all conditions, being much higher than those in the controls on all days. For example, on day 6, the cell numbers were increased by 85.92-fold in cultures with leucine (0.325 g/L), 48.27-fold in cultures with leucine (0.65 g/L), 37.82-fold in cultures with sodium glutamate (0.325 g/L), 174.38-fold in cultures with maltose (0.325 g/L), 42-fold in cultures with maltose (0.325 g/L), 75-fold in cultures with maltose (0.65 g/L), 58.77-fold in cultures with maltose (1.3 g/L), and 37.82-fold in cultures with maltose (2.6 g/L). The numbers were similarly high for the other days.

There were no differences in the cell number in cultures grown in JM (−N) supplemented with various organic compounds compared to that of JM (+N), with the exception of a 51.35-fold increase on day 6 with sodium glutamate (0.325 g/L), increases of 50.65-fold and 59.35-fold on day 9 with leucine (0.65 g/L) and sodium glutamate (0.325 g/L), respectively, and increases of 64.65-fold, 104.65-fold, 126.15-fold, and 58.35-fold on day 12 with lysine (0.325 g/L), leucine (0.65 g/L), leucine (1.3 g/L), and sodium glutamate (0.325 g/L), respectively. *H. lacustris* grown in JM (+N) and JM (−N) supplemented with benzoic acid and malic acid at all concentrations died after 3 days of culture.

Astaxanthin extraction and estimation were performed with 14-day cultures of *H. lacustris* grown in JM (+N) and JM (–N) supplemented with nine organic compounds at concentrations of 0.325 g/L, 0.65 g/J, 1.3 g/L, and 2.6 g/L (Table 2, Table 3, Table 4, Table 5, Table 6, Table 7, Table 8, Table 9, Table 10, Table 11, Table 12, Table 13, Table 14, Table 15, Table 16, Table 17, Table 18 and Table 19). When grown in JM (−N), the astaxanthin contents were higher than those in controls with almost all organic compounds except cellulose (0.325 g/L), malic acid (0.325 g/L, 0.65 g/L, 1.325 g/L, 2.6 g/L), maltose (0.325 g/L, 0.65 g/L, 1.325 g/L, 2.6 g/L), and benzoic acid (0.325 g/L, 0.65 g/L, 1.325 g/L, 2.6 g/L). The levels were higher than 6 mg/L when grown with alanine (0.65 g/L), alanine (1.3 g/L), lysine (2.6 g/L), leucine (0.325 g/L), leucine (0.65 g/L), glutamine (0.325 g/L), glutamine (0.65 g/L), sodium glutamate (0.325 g/L), sodium glutamate (0.65 g/L), and sodium glutamate (1.3 g/L); the highest levels of 9.099 mg/L were observed in the presence of leucine at 0.65 g/L.

In this experiment, astaxanthin levels were higher after growth in JM (+N) than those after growth in JM (−N). The levels grown with lysine (1.3 g/L), sodium glutamate (0.325 g/L), cellulose (0.325 g/L), cellulose (0.65 g/L), and cellulose (2.6 g/L) were higher than the control level of 11.637 mg/L, and the highest astaxanthin content of 20.484 mg/L was observed when grown in the presence of sodium glutamate at 0.325 g/L.

## 4. Discussion

Cultivation under conditions of nitrogen starvation is considered necessary to produce astaxanthin from *H. lacustris*. As JM showed the highest concentration of astaxanthin (5.87 mg/mL) under nitrogen-depleted conditions, we found that JM (−N) was the optimal medium for astaxanthin production by *H. lacustris*. Moreover, another reason for selecting JM was its cost-effectiveness on a larger scale. As we can see, the concentrations of the chemical ingredients required for JM are lower. However, when we compared the cell numbers and astaxanthin production of *H. lacustris* during the organic compounds supplementation experiment with the media experiment, the results showed that the cell numbers and astaxanthin production of *H. lacustris* grown under JM (+N) were higher than those under JM (−N). This could be dependent on the initial cell number of *H. lacustris*. The initial cell numbers of *H. lacustris* grown under JM (+N) and JM (−N) for the media experiment were higher by 15.45-fold and 10.05-fold, respectively, than for the organic compounds supplementation experiment. Therefore, there is still a further requirement for additional experiments to optimize the initial cell numbers for the commercial production of astaxanthin from *H. lacustris*. Five of the nine organic carbon sources tested as supplements—alanine, leucine, lysine, glutamine, and sodium glutamate—contain nitrogen. Nitrogen has been shown to play an important role in both cell growth and astaxanthin accumulation by *H. lacustris* [39]. A previous study examined the effects of four organic carbon sources on astaxanthin production by *H. lacustris*, i.e., glycerol, sodium acetate, ribose, and sodium gluconate [5]. Nitrogen is a vital nutrient in the growth medium of *H. lacustris*. Its depletion induces stress responses, as a result of which the green motile cells stop dividing and turn into hematocysts ready for the accumulation of astaxanthin with the increased activity of the Tricarboxylic acid (TCA) cycle [40,41]. The range of optimal light intensity for microalgae is between 26 and 400 µmol photons m^−2^ s^−1^ [42]. Nitrogen plays an essential role in the cultivation of microalgae since it is a critical component of biological macromolecules such as DNA, protein, and chlorophyll. Nitrogen deficiency can enhance lipid accumulation and lower the biomass yield [43]. Astaxanthin accumulated in *H. lacustris* under growth-limiting conditions, such as high salinity, high light intensity, and nutrient deficiency. Nitrogen is also an essential nutrient responsible for enzymatic activities and the cell growth of *H. lacustris*. Its deficiency inhibits the biosynthesis of chlorophyll and thus impairs photosynthetic function [44]. Moreover, we also evaluated the effects of a low light intensity and lower temperature coupled with different media and supplementation with organic compounds on the growth and astaxanthin accumulation of *H. lacustris*. Cultures were illuminated with cool-white fluorescent lamps with an intensity of 40 µmol m^−2^ s^−1^ [12], and the experiments were performed at the relatively low temperature of 16 °C ± 2 °C.

When *H. lacustris* was grown in JM (+N) media supplemented with various organic compounds, the greatest increment in cell number (174.38-fold) was observed on day 6 in the culture with maltose (0.325 g/L). On day 9, the greatest increment in the cell number (129.76-fold) was again seen with maltose (0.325 g/L), while on day 12, the greatest increment in the cell number (115.65-fold) was seen with leucine (1.3 g/L). Similarly, when grown in JM (−N) supplemented with various organic compounds, the greatest increment in the cell number (51.35-fold) was seen with sodium glutamate (0.325 g/L) on day 6. The greatest increment in the cell number on day 9 (59.35-fold) was again seen with sodium glutamate (0.325 g/L), while on day 12, the greatest (126.15-fold) was seen with leucine (1.3 g/L). The results showed improvements in astaxanthin production by *H. lacustris* in JM (−N) supplemented with alanine (0.325 g/L, 0.65 g/L, 1.3 g/L), lysine (2.6 g/L), leucine (0.325 g/L, 0.65 g/L, 1.3 g/L, 2.6 g/L), glutamine (0.325 g/L, 0.65 g/L, 1.3 g/L), and sodium glutamate (0.325 g/L, 0.65 g/L, 1.3 g/L) compared to a previous study that supplemented the culture medium with ribose, sodium acetate, sodium gluconate, and glycerol [5]. Similarly, when cultivated in JM (+N) supplemented with various organic compounds, *H. lacustris* showed increases in astaxanthin contents of 1.17-fold in a medium supplemented with lysine (1.3 g/L), 1.75-fold in a medium supplemented with sodium glutamate (0.325 g/L), 1.52-fold in a medium supplemented with cellulose (0.325 g/L), 1.04-fold in a medium supplemented with cellulose (0.65 g/L), and 1.08-fold in a medium supplemented with cellulose (2.6 g/L) compared to the control. These observations suggest that astaxanthin production can be improved even without the removal of nitrogen sources from JM.

## 5. Conclusions

Culture in JM (−N) supplemented with leucine (0.65 g/L) and sodium glutamate (0.325 g/L) significantly enhanced the accumulation of astaxanthin in *H. lacustris*. Astaxanthin production was increased in JM (+N) supplemented with lysine (1.3 g/L), sodium glutamate (0.325 g/L), cellulose (0.325 g/L), cellulose (0.65 g/L), and cellulose (2.6 g/L). Similarly, when cultured in JM (−N), its production was increased by the supplementation of the medium with alanine (0.325 g/L, 0.65 g/L, 1.3 g/L), lysine (2.6 g/L), leucine (0.325 g/L, 0.65 g/L, 1.3 g/L, 2.6 g/L), glutamine (0.325 g/L, 0.65 g/L, 1.3 g/L), and sodium glutamate (0.325 g/L, 0.65 g/L, 1.3 g/L). Overall, the supplementation of four amino acids (glutamine, alanine, lysine, and leucine) along with sodium glutamate and cellulose enhanced the astaxanthin accumulation in *H. lacustris*. Glutamate feeding enhanced the yield of astaxanthin in *Xanthophyllomyces dendrorhous*. Apart from serving as a nitrogen source in a cell, it also participates in the TCA cycle [45]. There is a great deal of interest in astaxanthin production from microalgae due to the health benefits of natural astaxanthin. Further studies are required on the solubility of cellulose in JM and to determine the mechanisms underlying the effects of supplementation with lysine, leucine, glutamine, alanine, and cellulose on the growth and astaxanthin accumulation of *H. lacustris* at the molecular level.

## Figures and Tables

**Figure 1 microorganisms-12-01040-f001:**
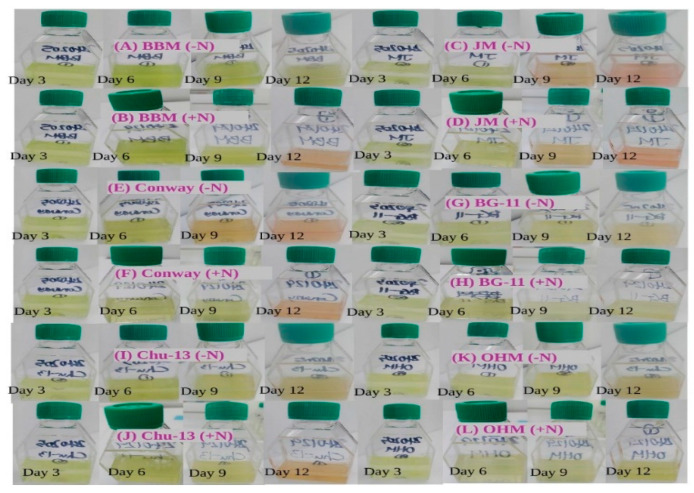
Appearances of *H. lacustris* cultures grown in different media (Day 3, 6, 9, and 12). (**A**) *H. lacustris* cultures grown in BBM (−N). (**B**) *H. lacustris* cultures grown in BBM (+N). (**C**) *H. lacustris* cultures grown in JM (−N). (**D**) *H. lacustris* cultures grown in JM (+N). (**E**) *H. lacustris* cultures grown in Conway (−N). (**F**) *H. lacustris* cultures grown in Conway (+N). (**G**) *H. lacustris* cultures grown in BG-11 (−N). (**H**) *H. lacustris* cultures grown in BG-11 (+N). (**I**) *H. lacustris* cultures grown in CHU-13 (−N). (**J**) *H. lacustris* cultures grown in CHU-13 (+N). (**K**) *H. lacustris* cultures grown in OHM (−N). (**L**) *H. lacustris* cultures grown in OHM (+N).

**Figure 2 microorganisms-12-01040-f002:**
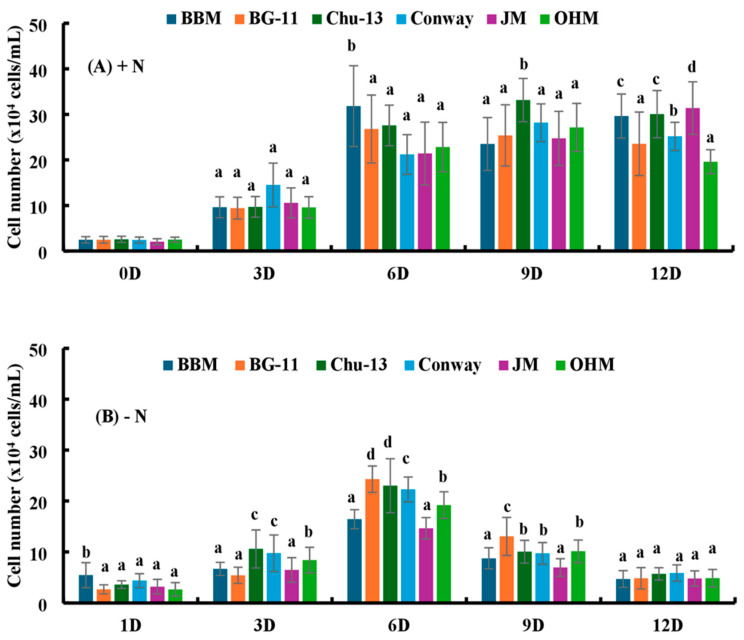
(**A**) Comparison of the cell number of *H. lacustris* under the influence of different media with nitrogen source (0D—Day 0, 3D—Day 3, 6D—Day 6, 9D—Day 9, 12D—Day 12). (**B**) Comparison of the cell number of *H. lacustris* under the influence of different media without nitrogen source (1D—Day 1, 3D—Day 3, 6D—Day 6, 9D—Day 9, 12D—Day 12). Mean values that do not share the same letter are significantly different at *p* < 0.001.

**Figure 3 microorganisms-12-01040-f003:**
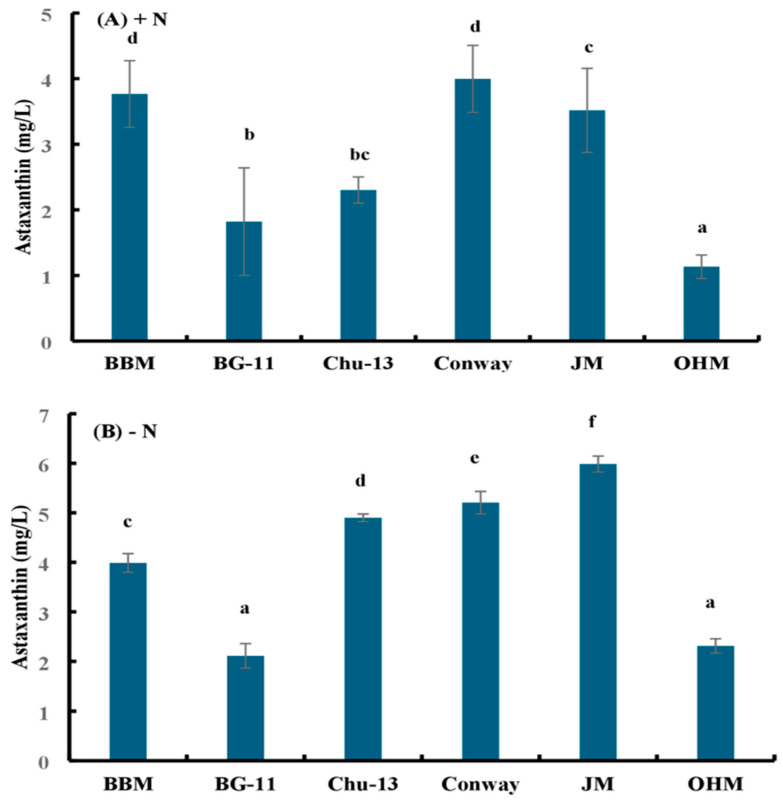
Astaxanthin accumulation by *H. lacustris* grown in different media in 14 days (**A**) with a nitrogen source and (**B**) without a nitrogen source. Mean values that do not share the same letter are significantly different at *p* < 0.001.

**Table 1 microorganisms-12-01040-t001:** Chemical composition of six different media used in this study.

Chemical	Walne/Conway	OHM	BG-11	Mod. CHU 13	BBM	JM
KNO_3_	100 g/L	410 mg/L	-	0.2 g/L	-	0.05 mg/L
NaNO_3_	-	-	1.5 g/L	-	1.5 g/L	0.08 mg/L
Ca(NO_3_)_2_·4H_2_O	-	-	-	-	-	0.02 mg/L
Ferric ammonium citrate	-	-	0.012 g/L	-	6 mg/L	-
Na_2_EDTA	-	-	0.001 g/L	-	1 mg/L	2.25 µg/L
K_2_HPO_4_	-	-	0.04 g/L	0.04 g/L	40 mg/L	0.0124 mg/L
Na_3_PO_4_	20 g/L	-	-	-	-	-
Na_2_HPO_4_	-	30 mg/L	-	-	-	0.036 mg/L
EDTAFeNa	-	-	-	-	-	2.25 µg/L
Na_2_CO_3_	-	-	0.02 g/L	-	-	-
MgSO_4_·7H_2_O	-	246.5 mg/L	36.1 mg/L	0.1 g/L	75 mg/L	-
CaCl_2_·2H_2_O	-	110.9 mg/L	27.2 mg/L	0.054 g/L	36 mg/L	-
Citric acid	-	-	-	100.1 mg/L	6 mg/L	-
Sodium citrate	-	-	8.82 mg/L	-	-	-
NaHCO_3_	-	-	0.036 g/L	-	-	15.9 µg/L
FeC_6_H_5_O7·5H_2_O	-	2.62 mg/L	-	0.01 g/L	-	-
Na_2_H_2_EDTA.2H_2_O	45 g/L	-	-	-	-	-
FeCl_3_·6H_2_O	1.3 g/L	-	-	-	-	-
ZnCl_2_	4.2 g/L	-	-	-	-	-
MnCl_2_·4H_2_O	0.36 g/L	0.989 mg/L	1.81 mg/L	1.8 mg/L	1.81 mg/L	0.139 µg/L
CoCl_2_·6H_2_O	4 g/L	0.011 mg/L	-	0.08 mg/L	-	-
CuSO_4_·5H_2_O	4 g/L	0.012 mg/L	0.079 mg/L	0.08 mg/L	0.08 mg/L	-
Cr_2_O_3_	-	0.076 mg/L	-	-	-	-
Na_2_MoO_4_·2H_2_O	-	0.12 mg/L	0.39 mg/L	0.05 mg/L	0.39 mg/L	-
(NH_4_)_6_Mo_7_O_24_·4H_2_O	1.8 g/L	-	-	-	-	0.001 mg/L
H_3_BO_3_	33.4 g/L	-	2.86 mg/L	2.85 mg/L	2.86 mg/L	2.48 µg/L
Thiamine—HCl	200 mg/L	-	-	-	-	0.04 µg/L
Cyanocobalamin	10 mg/L	15 µg/mL	-	-	-	0.04 µg/L
Biotin		25 µg/mL	--	-	-	0.04 µg/L
Thiamine	-	17.5 µg/mL	-	-	-	-
ZnSO_4_·7H_2_O	-	-	0.222 mg/L	0.02 mg/L	0.222 mg/L	-
Co(NO_3_)_2_·6H_2_O	-	-	0.049 mg/L	-	-	-

**Table 2 microorganisms-12-01040-t002:** Cell number of *H. lacustris* grown in JM (+N) media (Day 1, Day 3, Day 6, Day 9, Day 12) and astaxanthin contents of *H. lacustris* on Day 14 grown with the supplementation of benzoic acid. Data are presented as the mean ± standard deviation.

*H. lacustris* Culture Grown with the Supplementation of Benzoic Acid	Day 1 Cell Number (×10^4^ Cells/mL)	Day 3 Cell Number (×10^4^ Cells/mL)	Day 6 Cell Number (×10^4^ Cells/mL)	Day 9 Cell Number (×10^4^ Cells/mL)	Day 12 Cell Number (×10^4^ Cells/mL)	Day 14 Astaxanthin Content (mg/L)
Control	0.20 ± 0.000	3.20 ± 0.566	12.93 ± 1.320	11.47 ± 1.676	22.40 ± 5.571	11.64 ± 0.077
Benzoic acid (0.325 g/L)	0.20 ± 0.000	0.07 ± 0.094	0.00 ± 0.000	0.00 ± 0.000	0.00 ± 0.000	0.00 ± 0.000
Benzoic acid (0.65 g/L)	0.20 ± 0.000	0.00 ± 0.000	0.00 ± 0.000	0.00 ± 0.000	0.00 ± 0.000	0.00 ± 0.000
Benzoic acid (1.3 g/L)	0.13 ± 0.094	0.00 ± 0.000	0.00 ± 0.000	0.00 ± 0.000	0.00 ± 0.000	0.00 ± 0.000
Benzoic acid (2.6 g/L)	0.27 ± 0.249	0.00 ± 0.000	0.00 ± 0.000	0.00 ± 0.000	0.00 ± 0.000	0.00 ± 0.000

**Table 3 microorganisms-12-01040-t003:** Cell number of *H. lacustris* grown in JM (+N) media (Day 1, Day 3, Day 6, Day 9, Day 12) and astaxanthin contents of *H. lacustris* on Day 14 grown with the supplementation of cellulose. Data are presented as the mean ± standard deviation. Lowercase letters indicate significant differences (*p* < 0.001).

*H. lacustris* Culture Grown with the Supplementation of Cellulose	Day 1 Cell Number (×10^4^ Cells/mL)	Day 3 Cell Number (×10^4^ Cells/mL)	Day 6 Cell Number (×10^4^ Cells/mL)	Day 9 Cell Number (×10^4^ Cells/mL)	Day 12 Cell Number (×10^4^ Cells/mL)	Day 14 Astaxanthin Content (mg/L)
Control	0.20 ± 0.000	3.20 ± 0.566	12.93 ± 1.320 ^b^	11.47 ± 1.676	22.40 ± 5.571 ^ab^	11.64 ± 0.077 ^a^
Cellulose (0.325 g/L)	0.27 ± 0.094	3.93 ± 0.660	10.93 ± 1.268 ^ab^	14.47 ± 3.151	29.40 ± 0.432 ^b^	17.76 ± 0.067 ^d^
Cellulose (0.65 g/L)	0.40 ± 0.283	2.93 ± 0.411	9.93 ± 1.76 ^ab^	13.27 ± 2.131	24.33 ± 0.772 ^ab^	12.17 ± 0.017 ^b^
Cellulose (1.3 g/L)	0.27 ± 0.094	9.93 ± 1.761	8.33 ± 0.525 ^a^	12.40 ± 0.589	20.73 ± 0.340 ^a^	11.79 ± 0.048 ^ab^
Cellulose (2.6 g/L)	0.13 ± 0.094	2.47 ± 0.411	9.00 ± 0.993 ^ab^	13.20 ± 2.592	20.13 ± 0.838 ^a^	12.60 ± 0.294 ^c^

**Table 4 microorganisms-12-01040-t004:** Cell number of *H. lacustris* grown in JM (+N) media (Day 1, Day 3, Day 6, Day 9, Day 12) and astaxanthin contents of *H. lacustris* on Day 14 grown with the supplementation of glutamine. Data are presented as the mean ± standard deviation. Lowercase letters indicate significant differences (*p* < 0.001).

*H. lacustris* Culture Grown with the Supplementation of Glutamine	Day 1 Cell Number (×10^4^ Cells/mL)	Day 3 Cell Number (×10^4^ Cells/mL)	Day 6 Cell Number (×10^4^ Cells/mL)	Day 9 Cell Number (×10^4^ Cells/mL)	Day 12 Cell Number (×10^4^ Cells/mL)	Day 14 Astaxanthin Content (mg/L)
Control	0.20 ± 0.000	3.20 ± 0.566 ^b^	12.93 ± 1.320 ^c^	11.47 ± 1.676 ^c^	22.40 ± 5.571 ^c^	11.64 ± 0.077 ^e^
Glutamine (0.325 g/L)	0.47 ± 0.094	3.07 ± 0.822 ^b^	6.87 ± 1.389 ^b^	3.13 ± 0.736 ^a^	1.73 ± 0.411 ^b^	8.80 ± 0.106 ^d^
Glutamine (0.65 g/L)	0.60 ± 0.432	1.27 ± 0.189 ^a^	2.60 ± 0.589 ^a^	2.20 ± 0.909 ^a^	0.40 ± 0.163 ^a^	5.92 ± 0.006 ^c^
Glutamine (1.3 g/L)	0.33 ± 0.094	0.33 ± 0.094 ^a^	0.07 ± 0.094	0.07 ± 0.094	0.07 ± 0.094	1.31 ± 0.029 ^b^
Glutamine (2.6 g/L)	0.33 ± 0.094	0.73 ± 0.411 ^a^	0.00 ± 0.000	0.00 ± 0.000	0.00 ± 0.000	0.01 ± 0.003 ^a^

**Table 5 microorganisms-12-01040-t005:** Cell number of *H. lacustris* grown in JM (+N) media (Day 1, Day 3, Day 6, Day 9, Day 12) and astaxanthin contents of *H. lacustris* on Day 14 grown with the supplementation of alanine. Data are presented as the mean ± standard deviation. Lowercase letters indicate significant differences (*p* < 0.001).

*H. lacustris* Culture Grown with the Supplementation of Alanine	Day 1 Cell Number (×10^4^ Cells/mL)	Day 3 Cell Number (×10^4^ Cells/mL)	Day 6 Cell Number (×10^4^ Cells/mL)	Day 9 Cell Number (×10^4^ Cells/mL)	Day 12 Cell Number (×10^4^ Cells/mL)	Day 14 Astaxanthin Content (mg/L)
Control	0.20 ± 0.000	3.20 ± 0.566	12.93 ± 1.320 ^d^	11.47 ± 1.676 ^c^	22.40 ± 5.571 ^c^	11.64 ± 0.077 ^c^
Alanine (0.325 g/L)	0.33 ± 0.094	3.40 ± 0.864	10.27 ± 0.249 ^c^	5.27 ± 0.660 ^b^	4.60 ± 0.163 ^b^	7.03 ± 0.048 ^b^
Alanine (0.65 g/L)	0.47 ± 0.094	0.07 ± 0.094	3.00 ± 0.432 ^b^	3.00 ± 0.748 ^a^	0.73 ± 0.249 ^a^	7.05 ± 0.006 ^b^
Alanine (1.3 g/L)	0.13 ± 0.094	0.07 ± 0.094	0.07 ± 0.094 ^a^	0.00 ± 0.000	0.07 ± 0.094	1.34 ± 0.076 ^a^
Alanine (2.6 g/L)	0.40 ± 0.163	0.07 ± 0.094	0.00 ± 0.000	0.00 ± 0.000	0.00 ± 0.000	0.23 ± 0.017 ^a^

**Table 6 microorganisms-12-01040-t006:** Cell number of *H. lacustris* grown in JM (+N) media (Day 1, Day 3, Day 6, Day 9, Day 12) and astaxanthin contents of *H. lacustris* on Day 14 grown with the supplementation of leucine. Data are presented as the mean ± standard deviation. Lowercase letters indicate significant differences (*p* < 0.001).

*H. lacustris* Culture Grown with the Supplementation of Leucine	Day 1 Cell Number (×10^4^ Cells/mL)	Day 3 Cell Number (×10^4^ Cells/mL)	Day 6 Cell Number (×10^4^ Cells/mL)	Day 9 Cell Number (×10^4^ Cells/mL)	Day 12 Cell Number (×10^4^ Cells/mL)	Day 14 Astaxanthin Content (mg/L)
Control	0.20 ± 0.000	3.20 ± 0.566 ^ab^	12.93 ± 1.320 ^a^	11.47 ± 1.676 ^a^	22.40 ± 5.571	11.64 ± 0.077 ^e^
Leucine (0.325 g/L)	0.27 ± 0.094	3.47 ± 0.525 ^b^	23.20 ± 2.953 ^b^	23.27 ± 0.806 ^c^	21.40 ± 2.673	11.48 ± 0.073 ^d^
Leucine (0.65 g/L)	0.33 ± 0.094	2.93 ± 0.680 ^ab^	15.93 ± 1.746 ^a^	15.80 ± 1.766 ^b^	19.47 ± 1.636	7.33 ± 0.017 ^c^
Leucine (1.3 g/L)	0.20 ± 0.163	1.60 ± 0.163 ^a^	10.27 ± 1.310 ^a^	19.73 ± 0.411 ^c^	23.13 ± 3.465	5.33 ± 0.011 ^b^
Leucine (2.6 g/L)	0.27 ± 0.094	2.27 ± 0.618 ^ab^	12.33 ± 1.087 ^a^	15.07 ± 1.676 ^ab^	20.60 ± 0.748	5.18 ± 0.006 ^a^

**Table 7 microorganisms-12-01040-t007:** Cell number of *H. lacustris* grown in JM (+N) media (Day 1, Day 3, Day 6, Day 9, Day 12) and astaxanthin contents of *H. lacustris* on Day 14 grown with the supplementation of lysine. Data are presented as the mean ± standard deviation. Lowercase letters indicate significant differences (*p* < 0.001).

*H. lacustris* Culture Grown with the Supplementation of Lysine	Day 1 Cell Number (×10^4^ Cells/mL)	Day 3 Cell Number (×10^4^ Cells/mL)	Day 6 Cell Number (×10^4^ Cells/mL)	Day 9 Cell Number (×10^4^ Cells/mL)	Day 12 Cell Number (×10^4^ Cells/mL)	Day 14 Astaxanthin Content (mg/L)
Control	0.20 ± 0.000	3.20 ± 0.566 ^b^	12.93 ± 1.320 ^b^	11.47 ± 1.676 ^bc^	22.40 ± 5.571 ^bc^	11.64 ± 0.077 ^d^
Lysine (0.325 g/L)	0.27 ± 0.094	4.67 ± 1.087 ^b^	12.73 ± 0.411 ^b^	14.60 ± 0.589 ^c^	20.73 ± 2.205 ^bc^	0.04 ± 0.000 ^a^
Lysine (0.65 g/L)	0.80 ± 0.327	4.13 ± 0.806 ^b^	12.87 ± 1.668 ^b^	26.47 ± 3.356 ^d^	38.67 ± 11.927 ^c^	10.15 ± 0.006 ^c^
Lysine (1.3 g/L)	0.27 ± 0.094	1.07 ± 0.525 ^a^	1.93 ± 0.618 ^a^	7.00 ± 1.395 ^b^	15.27 ± 0.660 ^ab^	13.62 ± 0.061 ^e^
Lysine (2.6 g/L)	0.20 ± 0.000	0.07 ± 0.094 ^a^	0.13 ± 0.094 ^a^	0.53 ± 0.340 ^a^	1.00 ± 0.327 ^a^	7.65 ± 0.013 ^b^

**Table 8 microorganisms-12-01040-t008:** Cell number of *H. lacustris* grown in JM (+N) media (Day 1, Day 3, Day 6, Day 9, Day 12) and astaxanthin contents of *H. lacustris* on Day 14 grown with the supplementation of malic acid.

*H. lacustris* Culture Grown with the Supplementation of Malic Acid	Day 1 Cell Number (×10^4^ Cells/mL)	Day 3 Cell Number (×10^4^ Cells/mL)	Day 6 Cell Number (×10^4^ Cells/mL)	Day 9 Cell Number (×10^4^ Cells/mL)	Day 12 Cell Number (×10^4^ Cells/mL)	Day 14 Astaxanthin Content (mg/L)
Control	0.20 ± 0.000	3.20 ± 0.566	12.93 ± 1.320	11.47 ± 1.676	22.40 ± 5.571	11.64 ± 0.077
Malic acid (0.325 g/L)	0.47 ± 0.094	0.00 ± 0.000	0.00 ± 0.000	0.00 ± 0.000	0.00 ± 0.000	0.13 ± 0.006
Malic acid (0.65 g/L)	0.67 ± 0.094	0.00 ± 0.000	0.00 ± 0.000	0.00 ± 0.000	0.00 ± 0.000	0.23 ± 0.048
Malic acid (1.3 g/L)	0.13 ± 0.094	0.00 ± 0.000	0.00 ± 0.000	0.00 ± 0.000	0.00 ± 0.000	0.03 ± 0.006
Malic acid (2.6 g/L)	0.20 ± 0.163	0.00 ± 0.000	0.00 ± 0.000	0.00 ± 0.000	0.00 ± 0.000	0.00 ± 0.000

**Table 9 microorganisms-12-01040-t009:** Cell number of *H. lacustris* grown in JM (+N) media (Day 1, Day 3, Day 6, Day 9, Day 12) and astaxanthin contents of *H. lacustris* on Day 14 grown with the supplementation of maltose. Data are presented as the mean ± standard deviation. Lowercase letters indicate significant differences (*p* < 0.001).

*H. lacustris* Culture Grown with the Supplementation of Maltose	Day 1 Cell Number (×10^4^ Cells/mL)	Day 3 Cell Number (×10^4^ Cells/mL)	Day 6 Cell Number (×10^4^ Cells/mL)	Day 9 Cell Number (×10^4^ Cells/mL)	Day 12 Cell Number (×10^4^ Cells/mL)	Day 14 Astaxanthin Content (mg/L)
Control	0.20 ± 0.000	3.20 ± 0.566	12.93 ± 1.320 ^a^	11.47 ± 1.676	22.40 ± 5.571 ^b^	11.64 ± 0.077
Maltose (0.325 g/L)	0.13 ± 0.094	2.00 ± 0.163	22.67 ± 2.391 ^b^	16.87 ± 2.584	22.13 ± 4.620 ^b^	0.00 ± 0.000
Maltose (0.65 g/L)	0.40 ± 0.163	3.40 ± 0.712	16.80 ± 1.337 ^a^	13.40 ± 0.589	14.67 ± 1.236 ^ab^	0.00 ± 0.000
Maltose (1.3 g/L)	0.20 ± 0.163	3.33 ± 0.340	15.00 ± 2.007 ^a^	13.13 ± 0.680	9.60 ± 1.020 ^a^	0.00 ± 0.000
Maltose (2.6 g/L)	0.27 ± 0.094	3.60 ± 0.849	16.20 ± 2.068 ^a^	12.73 ± 2.553	11.87 ± 2.007 ^ab^	0.00 ± 0.000

**Table 10 microorganisms-12-01040-t010:** Cell number of *H. lacustris* grown in JM (+N) media (Day 1, Day 3, Day 6, Day 9, Day 12) and astaxanthin contents of *H. lacustris* on Day 14 grown with the supplementation of sodium glutamate. Data are presented as the mean ± standard deviation. Lowercase letters indicate significant differences (*p* < 0.001).

*H. lacustris* Culture Grown with the Supplementation of Sodium Glutamate	Day 1 Cell Number (×10^4^ Cells/mL)	Day 3 Cell Number (×10^4^ Cells/mL)	Day 6 Cell Number (×10^4^ Cells/mL)	Day 9 Cell Number (×10^4^ Cells/mL)	Day 12 Cell Number (×10^4^ Cells/mL)	Day 14 Astaxanthin Content (mg/L)
Control	0.20 ± 0.000	3.20 ± 0.566 ^a^	12.93 ± 1.320 ^c^	11.47 ± 1.676 ^c^	22.40 ± 5.571 ^b^	11.64 ± 0.077 ^d^
Sodium Glutamate (0.325 g/L)	0.40 ± 0.000	1.93 ± 0.340 ^c^	15.13 ± 0.998 ^c^	12.73 ± 0.411 ^c^	25.00 ± 2.592 ^b^	20.48 ± 0.017 ^e^
Sodium Glutamate (0.65 g/L)	0.33 ± 0.094	1.40 ± 0.163 ^bc^	6.87 ± 0.957 ^b^	5.53 ± 0.573 ^b^	5.27 ± 0.754 ^a^	8.30 ± 0.048 ^c^
Sodium Glutamate (1.3 g/L)	0.40 ± 0.163	0.80 ± 0.000 ^ab^	3.07 ± 0.943 ^a^	3.33 ± 0.411 ^b^	0.53 ± 0.094 ^a^	5.62 ± 0.011 ^b^
Sodium Glutamate (2.6 g/L)	0.53 ± 0.249	0.07 ± 0.094 ^a^	0.53 ± 0.189 ^a^	0.20 ± 0.000 ^a^	0.13 ± 0.094 ^a^	0.55 ± 0.019 ^a^

**Table 11 microorganisms-12-01040-t011:** Cell number of *H. lacustris* grown in JM (−N) media (Day 1, Day 3, Day 6, Day 9, Day 12) and astaxanthin contents of *H. lacustris* on Day 14 grown with the supplementation of benzoic acid. Data are presented as the mean ± standard deviation.

*H. lacustris* Culture Grown with the Supplementation of Benzoic Acid	Day 1 Cell Number (×10^4^ Cells/mL)	Day 3 Cell Number (×10^4^ Cells/mL)	Day 6 Cell Number (×10^4^ Cells/mL)	Day 9 Cell Number (×10^4^ Cells/mL)	Day 12 Cell Number (×10^4^ Cells/mL)	Day 14 Astaxanthin Content (mg/L)
Control	0.20 ± 0.163	0.20 ± 0.163	0.53 ± 0.189	0.87 ± 0.094	0.80 ± 0.163	0.91 ± 0.035
Benzoic acid (0.325 g/L)	0.13 ± 0.094	0.13 ± 0.094	0.20 ± 0.163	0.07 ± 0.094	0.00 ± 0.000	0.00 ± 0.000
Benzoic acid (0.65 g/L)	0.20 ± 0.163	0.07 ± 0.094	0.13 ± 0.094	0.13 ± 0.094	0.00 ± 0.000	0.00 ± 0.000
Benzoic acid (1.3 g/L)	0.20 ± 0.163	0.27 ± 0.249	0.13 ± 0.189	0.13 ± 0.189	0.00 ± 0.000	0.00 ± 0.000
Benzoic acid (2.6 g/L)	0.20 ± 0.000	0.13 ± 0.094	0.00 ± 0.000	0.00 ± 0.000	0.00 ± 0.000	0.00 ± 0.000

**Table 12 microorganisms-12-01040-t012:** Cell number of *H. lacustris* grown in JM (−N) media (Day 1, Day 3, Day 6, Day 9, Day 12) and astaxanthin contents of *H. lacustris* on Day 14 grown with the supplementation of cellulose. Data are presented as the mean ± standard deviation. Lowercase letters indicate significant differences (*p* < 0.001).

*H. lacustris* Culture Grown with the Supplementation of Cellulose	Day 1 Cell Number (×10^4^ Cells/mL)	Day 3 Cell Number (×10^4^ Cells/mL)	Day 6 Cell Number (×10^4^ Cells/mL)	Day 9 Cell Number (×10^4^ Cells/mL)	Day 12 Cell Number (×10^4^ Cells/mL)	Day 14 Astaxanthin Content (mg/L)
Control	0.20 ± 0.163	0.20 ± 0.163	0.53 ± 0.189 ^a^	0.87 ± 0.094	0.80 ± 0.163	0.91 ± 0.035 ^a^
Cellulose (0.325 g/L)	0.27 ± 0.094	0.47 ± 0.249	0.67 ± 0.094 ^a^	1.07 ± 0.094	1.47 ± 0.822	0.90 ± 0.006 ^a^
Cellulose (0.65 g/L)	0.20 ± 0.000	0.40 ± 0.163	0.73 ± 0.094 ^a^	0.93 ± 0.189	1.00 ± 0.589	1.04 ± 0.028 ^b^
Cellulose (1.3 g/L)	0.07 ± 0.094	0.47 ± 0.249	1.38 ± 0.189 ^b^	1.00 ± 0.163	1.13 ± 0.377	1.40 ± 0.023 ^c^
Cellulose (2.6 g/L)	0.07 ± 0.094	0.53 ± 0.340	1.47 ± 0.189 ^b^	1.87 ± 0.660	2.13 ± 0.525	2.02 ± 0.017 ^d^

**Table 13 microorganisms-12-01040-t013:** Cell number of *H. lacustris* grown in JM (−N) media (Day 1, Day 3, Day 6, Day 9, Day 12) and astaxanthin contents of *H. lacustris* on Day 14 grown with the supplementation of glutamine. Data are presented as the mean ± standard deviation. Lowercase letters indicate significant differences (*p* < 0.001).

*H. lacustris* Culture Grown with the Supplementation of Glutamine	Day 1 Cell Number (×10^4^ Cells/mL)	Day 3 Cell Number (×10^4^ Cells/mL)	Day 6 Cell Number (×10^4^ Cells/mL)	Day 9 Cell Number (×10^4^ Cells/mL)	Day 12 Cell Number (×10^4^ Cells/mL)	Day 14 Astaxanthin Content (mg/L)
Control	0.20 ± 0.163	0.20 ± 0.163 ^a^	0.53 ± 0.189 ^ab^	0.87 ± 0.094 ^c^	0.80 ± 0.163 ^b^	0.91 ± 0.035 ^a^
Glutamine (0.325 g/L)	0.20 ± 0.000	1.20 ± 0.163 ^b^	1.40 ± 0.283 ^c^	0.53 ± 0.189 ^b^	0.13 ± 0.189 ^a^	8.60 ± 0.046 ^e^
Glutamine (0.65 g/L)	0.13 ± 0.094	0.13 ± 0.094 ^a^	0.87 ± 0.094 ^bc^	0.13 ± 0.094 ^a^	0.00 ± 0.000	7.61 ± 0.011 ^d^
Glutamine (1.3 g/L)	0.07 ± 0.094	0.20 ± 0.163 ^a^	0.67 ± 0.094 ^ab^	0.07 ± 0.094 ^a^	0.00 ± 0.000	4.86 ± 0.006 ^c^
Glutamine (2.6 g/L)	0.13 ± 0.094	0.33 ± 0.094 ^a^	0.20 ± 0.283 ^a^	0.00 ± 0.000	0.00 ± 0.000	1.63 ± 0.025 ^b^

**Table 14 microorganisms-12-01040-t014:** Cell number of *H. lacustris* grown in JM (−N) media (Day 1, Day 3, Day 6, Day 9, Day 12) and astaxanthin contents of *H. lacustris* on Day 14 grown with the supplementation of alanine. Data are presented as the mean ± standard deviation. Lowercase letters indicate significant differences (*p* < 0.001).

*H. lacustris* Culture Grown with the Supplementation of Alanine	Day 1 Cell Number (×10^4^ Cells/mL)	Day 3 Cell Number (×10^4^ Cells/mL)	Day 6 Cell Number (×10^4^ Cells/mL)	Day 9 Cell Number (×10^4^ Cells/mL)	Day 12 Cell Number (×10^4^ Cells/mL)	Day 14 Astaxanthin Content (mg/L)
Control	0.20 ± 0.163	0.20 ± 0.163 ^a^	0.53 ± 0.189	0.87 ± 0.094	0.80 ± 0.163	0.91 ± 0.035 ^a^
Alanine (0.325 g/L)	0.20 ± 0.000	1.20 ± 0.163 ^b^	1.60 ± 0.589	1.20 ± 0.566	1.07 ± 0.471	5.76 ± 0.011 ^c^
Alanine (0.65 g/L)	0.27 ± 0.094	0.27 ± 0.094 ^a^	1.67 ± 0.499	1.00 ± 0.163	0.87 ± 0.094	6.06 ± 0.092 ^d^
Alanine (1.3 g/L)	0.07 ± 0.094	0.40 ± 0.163 ^a^	0.60 ± 0.163	0.87 ± 0.094	1.93 ± 0.525	6.13 ± 0.011 ^d^
Alanine (2.6 g/L)	0.20 ± 0.000	0.53 ± 0.499 ^ab^	0.53 ± 0.189	0.87 ± 0.094	1.47 ± 0.377	1.80 ± 0.013 ^b^

**Table 15 microorganisms-12-01040-t015:** Cell number of *H. lacustris* grown in JM (−N) media (Day 1, Day 3, Day 6, Day 9, Day 12) and astaxanthin contents of *H. lacustris* on Day 14 grown with the supplementation of leucine. Data are presented as the mean ± standard deviation. Lowercase letters indicate significant differences (*p* < 0.001).

*H. lacustris* Culture Grown with the Supplementation of Leucine	Day 1 Cell Number (×10^4^ Cells/mL)	Day 3 Cell Number (×10^4^ Cells/mL)	Day 6 Cell Number (×10^4^ Cells/mL)	Day 9 Cell Number (×10^4^ Cells/mL)	Day 12 Cell Number (×10^4^ Cells/mL)	Day 14 Astaxanthin Content (mg/L)
Control	0.20 ± 0.163	0.20 ± 0.163 ^a^	0.53 ± 0.189 ^a^	0.87 ± 0.094 ^a^	0.80 ± 0.163 ^a^	0.91 ± 0.035 ^a^
Leucine (0.325 g/L)	0.07 ± 0.094	0.93 ± 0.189 ^b^	1.73 ± 0.411 ^b^	3.13 ± 0.660 ^ab^	10.80 ± 0.589 ^c^	8.30 ± 0.090 ^d^
Leucine (0.65 g/L)	0.20 ± 0.000	0.67 ± 0.094 ^ab^	2.47 ± 0.618 ^bc^	10.13 ± 0.618 ^bc^	20.93 ± 2.614 ^e^	9.10 ± 0.029 ^e^
Leucine (1.3 g/L)	0.13 ± 0.094	0.47 ± 0.094 ^ab^	3.27 ± 0.249 ^c^	5.93 ± 1.112 ^b^	16.40 ± 0.993 ^d^	5.82 ± 0.032 ^c^
Leucine (2.6 g/L)	0.07 ± 0.094	0.53 ± 0.189 ^ab^	2.73 ± 0.499 ^bc^	4.93 ± 1.543 ^b^	7.07 ± 0.660 ^b^	2.17 ± 0.011 ^b^

**Table 16 microorganisms-12-01040-t016:** Cell number of *H. lacustris* grown in JM (−N) media (Day 1, Day 3, Day 6, Day 9, Day 12) and astaxanthin contents of *H. lacustris* on Day 14 grown with the supplementation of lysine. Data are presented as the mean ± standard deviation. Lowercase letters indicate significant differences (*p* < 0.001).

*H. lacustris* Culture Grown with the Supplementation of Lysine	Day 1 Cell Number (×10^4^ Cells/mL)	Day 3 Cell Number (×10^4^ Cells/mL)	Day 6 Cell Number (×10^4^ Cells/mL)	Day 9 Cell Number (×10^4^ Cells/mL)	Day 12 Cell Number (×10^4^ Cells/mL)	Day 14 Astaxanthin Content (mg/L)
Control	0.20 ± 0.163	0.20 ± 0.163	0.53 ± 0.189 ^a^	0.87 ± 0.094 ^ab^	0.80 ± 0.163 ^a^	0.91 ± 0.035 ^a^
Lysine (0.325 g/L)	0.20 ± 0.000	0.40 ± 0.163	1.47 ± 0.249 ^b^	7.53 ± 0.499 ^c^	12.93 ± 1.236 ^c^	1.99 ± 0.061 ^c^
Lysine (0.65 g/L)	0.07 ± 0.094	0.20 ± 0.163	0.93 ± 0.411 ^ab^	1.67 ± 0.499 ^b^	3.73 ± 0.499 ^b^	2.12 ± 0.006 ^d^
Lysine (1.3 g/L)	0.07 ± 0.094	0.20 ± 0.163	0.27 ± 0.377 ^a^	0.07 ± 0.094 ^a^	0.07 ± 0.094	1.85 ± 0.000 ^b^
Lysine (2.6 g/L)	0.13 ± 0.094	0.07 ± 0.094	0.13 ± 0.189 ^a^	0.00 ± 0.000	0.00 ± 0.000	7.36 ± 0.023 ^e^

**Table 17 microorganisms-12-01040-t017:** Cell number of *H. lacustris* grown in JM (−N) media (Day 1, Day 3, Day 6, Day 9, Day 12) and astaxanthin contents of *H. lacustris* on Day 14 grown with the supplementation of malic acid. Data are presented as the mean ± standard deviation. Lowercase letters indicate significant differences (*p* < 0.001).

*H. lacustris* Culture Grown with the Supplementation of Malic Acid	Day 1 Cell Number (×10^4^ Cells/mL)	Day 3 Cell Number (×10^4^ Cells/mL)	Day 6 Cell Number (×10^4^ Cells/mL)	Day 9 Cell Number (×10^4^ Cells/mL)	Day 12 Cell Number (×10^4^ Cells/mL)	Day 14 Astaxanthin Content (mg/L)
Control	0.20 ± 0.163	0.20 ± 0.163	0.53 ± 0.189	0.87 ± 0.094	0.80 ± 0.163	0.91 ± 0.035 ^c^
Malic acid (0.325 g/L)	0.07 ± 0.094	0.07 ± 0.094	1.27 ± 0.189	0.40 ± 0.327	0.00 ± 0.000	0.07 ± 0.000 ^a^
Malic acid (0.65 g/L)	0.20 ± 0.000	0.00 ± 0.000	1.00 ± 0.432	0.13 ± 0.094	0.00 ± 0.000	0.00 ± 0.000
Malic acid (1.3 g/L)	0.13 ± 0.094	0.00 ± 0.000	0.67 ± 0.340	0.07 ± 0.094	0.00 ± 0.000	0.00 ± 0.000
Malic acid (2.6 g/L)	0.13 ± 0.094	0.00 ± 0.000	0.93 ± 0.411	0.07 ± 0.094	0.00 ± 0.000	0.89 ± 0.055 ^b^

**Table 18 microorganisms-12-01040-t018:** Cell number of *H. lacustris* grown in JM (−N) media (Day 1, Day 3, Day 6, Day 9, Day 12) and astaxanthin contents of *H. lacustris* on Day 14 grown with the supplementation of maltose. Data are presented as the mean ± standard deviation. Lowercase letters indicate significant differences (*p* < 0.001).

*H. lacustris* Culture Grown with the Supplementation of Maltose	Day 1 Cell Number (×10^4^ Cells/mL)	Day 3 Cell Number (×10^4^ Cells/mL)	Day 6 Cell Number (×10^4^ Cells/mL)	Day 9 Cell Number (×10^4^ Cells/mL)	Day 12 Cell Number (×10^4^ Cells/mL)	Day 14 Astaxanthin Content (mg/L)
Control	0.20 ± 0.163	0.20 ± 0.163 ^a^	0.53 ± 0.189	0.87 ± 0.094 ^ab^	0.80 ± 0.163	0.91 ± 0.035 ^a^
Maltose (0.325 g/L)	0.13 ± 0.094	0.47 ± 0.094 ^ab^	0.87 ± 0.249	0.47 ± 0.189 ^a^	0.73 ± 0.094	0.89 ± 0.055 ^a^
Maltose (0.65 g/L)	0.20 ± 0.000	0.67 ± 0.189 ^ab^	1.00 ± 0.327	0.60 ± 0.283 ^a^	0.73 ± 0.094	0.73 ± 0.000 ^b^
Maltose (1.3 g/L)	0.13 ± 0.094	1.20 ± 0.566 ^b^	1.27 ± 0.499	1.33 ± 0.249 ^b^	1.27 ± 1.087	0.70 ± 0.006 ^ab^
Maltose (2.6 g/L)	0.20 ± 0.000	0.80 ± 0.189 ^ab^	0.87 ± 0.189	1.13 ± 0.094 ^ab^	0.60 ± 0.094	0.62 ± 0.046 ^a^

**Table 19 microorganisms-12-01040-t019:** Cell number of *H. lacustris* grown in JM (−N) media (Day 1, Day 3, Day 6, Day 9, Day 12) and astaxanthin contents of *H. lacustris* on Day 14 grown with the supplementation of sodium glutamate. Data are presented as the mean ± standard deviation. Lowercase letters indicate significant differences (*p* < 0.001).

*H. lacustris* Culture Grown with the Supplementation of Sodium Glutamate	Day 1 Cell Number (×10^4^ Cells/mL)	Day 3 Cell Number (×10^4^ Cells/mL)	Day 6 Cell Number (×10^4^ Cells/mL)	Day 9 Cell Number (×10^4^ Cells/mL)	Day 12 Cell Number (×10^4^ Cells/mL)	Day 14 Astaxanthin Content (mg/L)
Control	0.20 ± 0.163	0.20 ± 0.163 ^a^	0.53 ± 0.189 ^a^	0.87 ± 0.094 ^a^	0.80 ± 0.163 ^a^	0.91 ± 0.035 ^a^
Sodium Glutamate (0.325 g/L)	0.20 ± 0.000	1.27 ± 0.340 ^b^	10.27 ± 0.618 ^c^	11.87 ± 1.087 ^c^	11.67 ± 0.525 ^b^	6.03 ± 0.017 ^c^
Sodium Glutamate (0.65 g/L)	0.13 ± 0.094	1.27 ± 0.249 ^b^	2.87 ± 0.499 ^c^	1.93 ± 0.411 ^b^	0.73 ± 0.189 ^a^	8.01 ± 0.092 ^d^
Sodium Glutamate (1.3 g/L)	0.13 ± 0.094	0.33 ± 0.094 ^a^	0.33 ± 0.094 ^a^	0.20 ± 0.000 ^a^	0.07 ± 0.094 ^a^	8.45 ± 0.017 ^e^
Sodium Glutamate (2.6 g/L)	0.07 ± 0.094	0.33 ± 0.340 ^a^	0.27 ± 0.094 ^a^	0.27 ± 0.094 ^a^	0.13 ± 0.189 ^a^	2.00 ± 0.017 ^b^

## Data Availability

Data are contained within the article.

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
