# Peer review of "A Study on the Effect of Various Media and the Supplementation of Organic Compounds on the Enhanced Production of Astaxanthin from *Haematococcus lacustris* (Girod—Chantrans) Rostafinski (Chlorophyta)"

_microorganisms, 2024, doi:10.3390/microorganisms12061040_

Round 1
Reviewer 1 Report
Comments and Suggestions for Authors
Comments on microorganisms-2992580 entitled “A study on the effect of various media and supplementation of organic compounds on the enhanced production of astaxanthin from H. lacustris”
Research is devoted to interesting problem – effect of various media and supplementation of organic compounds on the enhanced production of astaxanthin from Haematococcus lacustris (Girod-Chantrans) Rostafinski. There is a high demand for natural astaxanthin as it possesses multiple health benefits. That is why the research is relevant.
Although researches on the production of astaxanthin using various groups of organisms is widespread, the study of most of investigated in the article organic compounds on the effect of astaxanthin production in H. lacustris has not been reported to date by any other research team.
The experimental methods and design are reasonable. The results are well-presented and have been carried out using different methods, including statistical analysis.
The paper is well written, and the text is clear and easy to read. I have only some remarks dealing with this manuscript:
1) Lines 2-4, “A study on the effect of various media and supplementation of organic compounds on the enhanced production of astaxanthin from H. lacustris” should be corrected to “A study on the effect of various media and supplementation of organic compounds on the enhanced production of astaxanthin from Haematococcus lacustris (Girod-Chantrans) Rostafinski (Chlorophyta)”.
2) Lines 22-23. “H. lacustris” should be corrected to “Haematococcus lacustris”.
3) Lines 42-43. “H. lacustris” should be corrected to “Haematococcus lacustris (Girod-Chantrans) Rostafinski (Chlorophyta)”.
4) Line 45. “Haematococcus lacustris” should be corrected to “H. lacustris”.
5) Line 101. “Spiriluna maxima” (not Spiriluna maxima, but Spirulina maxima (Setchell & N.L.Gardner) Geitler) should be corrected to “Limnospira maxima (Setchell & N.L.Gardner) Nowicka-Krawczyk, Mühlsteinová & Hauer”, according algaebase.org (Guiry, 2024).
6) Lines 110-112. “Chlorella pyrenoidosa” should be corrected to “Auxenochlorella pyrenoidosa (H.Chick) Molinari & Calvo-Pérez”, according algaebase.org (Guiry, 2024).
Author Response
- Lines 2-4, “A study on the effect of various media and supplementation of organic compounds on the enhanced production of astaxanthin from lacustris” should be corrected to “A study on the effect of various media and supplementation of organic compounds on the enhanced production of astaxanthin from Haematococcus lacustris (Girod-Chantrans) Rostafinski (Chlorophyta)”.
Answer: Thank you for the comment. The title has been changed to “A study on the effect of various media and supplementation of organic compounds on the enhanced production of astaxanthin from Haematococcus lacustris (Girod-Chantrans) Rostafinski (Chlorophyta)” as suggested. The changes are indicated by green color.
- Lines 22-23. “ lacustris” should be corrected to “Haematococcus lacustris”.
Answer: Thank you for the comment. The change has been made as suggested. The change is indicated by green color.
- Lines 42-43. “ lacustris” should be corrected to “Haematococcus lacustris (Girod-Chantrans) Rostafinski (Chlorophyta)”.
Answer: Thank you for the comment. The changes have been made as suggested. The changes are indicated by green color.
- Line 45. “Haematococcus lacustris” should be corrected to “ lacustris”.
Answer: Thank you for the comment. The change has been made as suggested. The change is indicated by green color.
- Line 101. “Spiriluna maxima” (not Spiriluna maxima, but Spirulina maxima(Setchell & N.L.Gardner) Geitler) should be corrected to “Limnospira maxima (Setchell & N.L.Gardner) Nowicka-Krawczyk, Mühlsteinová & Hauer”, according algaebase.org (Guiry, 2024).
Answer: Thank you for the comment. We have corrected this adding a new reference: Ghanchivahora, S.S. Evaluation of Limnospira Maxima Biomass and Microwave-assisted Extracts as Biofertilisers to Support Growth of Sorghum Bicolor; Flinders University, College of Medicine and Public Health, 2022, pp. 8. [CrossRef]. The changes are indicated by green color.
- Lines 110-112. “Chlorella pyrenoidosa” should be corrected to “Auxenochlorella pyrenoidosa (H.Chick) Molinari & Calvo-Pérez”, according algaebase.org (Guiry, 2024).
Answer: Thank you for the comment. We have corrected this adding a new reference: Chen, Z.; Su, B. Influence of Medium Frequency Light/Dark Cycles on the Cultivation of Auxenochlorella pyrenoidosa. Appl. Sci. 2020, 10, 5093. [CrossRef]. The changes are indicated by green color.
Reviewer 2 Report
Comments and Suggestions for Authors
The possibility of astaxanthin production by Haematococcus sp. has been the subject of numerous studies. Different types of media have been compared as a basis for algae cultivation. I believe that it is necessary to demonstrate what new this part of the research brings to this topic. Please explain why the JM medium that has been indicated as the most effective has the optimal composition and what determines it. Such information will be valuable for practical application. Why did the authors not refer to the media used under technical conditions. Since it has already been decided in practice which medium to use, why return to these considerations? For research purposes, this is worth clarifying.
Figure 1 does not provide any practical, measurable information, it is just a colorful decoration. Figures 2 - 4 also have no measurable value, and the magnification used does not allow any assessment of cell color.
There is no information on why certain organic compounds had a positive effect on astaxanthin production. The authors should try to provide information on why and how the presence of certain organic compounds affects the production of astaxanthin.
Author Response
- The possibility of astaxanthin production by Haematococcus sp. has been the subject of numerous studies. Different types of media have been compared as a basis for algae cultivation. I believe that it is necessary to demonstrate what new this part of the research brings to this topic. Please explain why the JM medium that has been indicated as the most effective has the optimal composition and what determines it. Such information will be valuable for practical application. Why did the authors not refer to the media used under technical conditions. Since it has already been decided in practice which medium to use, why return to these considerations? For research purposes, this is worth clarifying.
Answer: Thank you for the comment. Several new literatures have been added throughout the manuscript. The changes are indicated by green color.
- Figure 1 does not provide any practical, measurable information, it is just a colorful decoration. Figures 2 - 4 also have no measurable value, and the magnification used does not allow any assessment of cell color.
Answer: Thank you for the comment. We have retained figure 1 because we are describing the color changes of culture in the text but removed fig 2-4.
- There is no information on why certain organic compounds had a positive effect on astaxanthin production. The authors should try to provide information on why and how the presence of certain organic compounds affects the production of astaxanthin.
Answer: Thank you for the comment. We have added additional literature on the conclusion section. The changes are indicated by green color.
Reviewer 3 Report
Comments and Suggestions for Authors
REVIEW OF THE ARTICLE BY VIJAY RAYAMAJHI ET AL. ENTITLED “A STUDY ON THE EFFECT OF VARIOUS MEDIA AND SUPPLEMENTATION OF ORGANIC COMPOUNDS ON THE ENHANCED PRODUCTION OF ASTAXANTHIN FROM H. LACUSTRIS”
Rayamajhi et al. studied an effect of different organic sources on growth and astaxanthin accumulation in the commercially valuable chlorophyte Haematococcus lacustris (Chlamydomonadales): malic acid, benzoic acid, maltose, leucine, lysine, alanine, sodium glutamate, glutamine, and cellulose. Astaxanthin is a very important pigment widely used in cosmetics and aquaculture. I consider the data on the effect of different organic sources for mixotrophic growth of Haematococcus to be valuable for further research, making the topic relevant. The authors made a lot of experimental work. At the same time, the presentation of data and overall writing are very poor. The number of references (28) is extremely low for the Microbiology field. Some reverences to recent reviews in the field can be added. Therefore, the text should undergo a substantial revision. Please, see specific and general comments below.
TITLE
-Avoid shortenings in the title (H. lacustris should be Haematococcus lacustris).
ABSTRACT
-l. 22. Write the name in full at the first mention.
-l. 26. Nitrogen is a gas not fixed by chlorophytes. You should write “nitrogen source”.
-l. 24-29. Although selection of an optimal mineral medium is an important part of each algal work, it is routine and not novel. It diverts a reader from the main goal of the work and shades the novelty.I suggest focusing on the addition of the organic sources in the abstract and reducing the information about the mineral media.
INTRODUCTION
-Introduction is poorly written. The information close-related to the topic is not provided, despite such works exist. Previous data on H. lacustris growth on different organic sources (other than acetate) must be elucidated. See e.g. works doi.org/10.3390/life14010029, doi.org/10.1016/j.algal.2019.101779, doi.org/10.1016/j.algal.2019.101497, doi.org/10.1016/j.biortech.2021.125788, doi.org/10.1016/j.aquaculture.2021.736978, doi.org/10.1016/j.biortech.2020.124421, https://doi.org/10.1016/j.biortech.2019.121334, doi.org/10.1016/j.biortech.2021.126484.
-l. 39-44. References are missing.
-l. 47. “40-carbon” and “tetraterpene” are the same.
-l. 48. “linked with isoprene units” - cannot understand this. Astaxanthin is not linked with isoprene units. It consists of eight isoprene units as a structural basis.
-l. 48. “β rings” - should be “β-ionone rings” (see doi.org/10.3390/md21020108).
-l. 49. It is wrong. Astaxanthin has 13 double bonds in its π-conjugated system.
-l. 50-53. References are required.
-l. 56-63. The explanation of the Haematococcus life cycle should be revised. Motile cells are spores of asexual reproduction (zoospores). Spherical palmelloid vegetative cells can divide and turn to red aplanospores independently on the zoospore formation. You should also mention under which conditions astaxanthin accumulation is observed.
-l. 64-76. This cannot be the aim of a novel scientific study, since it has been done many times previously. This text is more suitable for methods.
-l. 82-83. “Four of the selected organic compounds: lysine, leucine, alanine, and glutamine are amino acids” - obvious textbook knowledge! remove.
-l. 92-94. Cannot understand the idea of this statement. What do you mean? In all living things growth is influenced by carbon metabolism. It is obvious.
-Correct name of Spirulina maxima is Limnospira maxima, and Chlorella pyrenoidosa is actually Auxenochlorella pyrenoidosa.
-l. 64-88. The goal of the study should be described at the end of the introduction.
-l. 129. It is obvious. sounds strange.
-l. 132. “The pH was adjusted to 7” - why? For example, in the original BG-11 protocol pH is 7.5.
-l. 133. “10 ml of cells” - what does it mean? How many cells?
-l. 136, 146. How was it measured? At which wavelength, how long was the optical path length? Which model of spectrophotometer was used?
-l. 139. Lux are not the units of photon flux density, they are the units of illumination. Photon flux density is 40 µmol m-2 s−1. Furthermore, the common spelling of lux is lx. You can write “illumination of 3,000 lx, which was equivalent to the photon flux density of 40 µmol m-2 s−1.”
-l. 149-153. Why are only these compounds listed here? There are also a set of mineral salts and vitamins.
-It is interesting for me, how you added cellulose to your media. It is a water-insoluble compound, so cannot be added to a homogeneous liquid system.
-l. 155. What is cell density? What is the difference between cell density and cell number?
-Table 1. Why did you add water-free Fe-citrate in one case and the pentahydrate in another?
-Table 1. CO should be Co.
-Table 1. Thiamin HCl should be Thiamine-HCl.
-Table 1. What is the difference between B12 and cyanocobalamin? Cyanocobalamin is a form of B12.
-Table 1. What is the difference between Ferric ammonium citrate and FeC6H5O7·NH4OH?
RESULTS
-l. 178, 184. “H. lacustris” should be italicised.
-l. 187-189. If you present this data, it should be properly described. In a separate paragraph panels on the figures should be explained. Please, specify which morphological changes you observe. I do not see any obvious changes on the figures.
-Figure 2-4: resolution of micrographs is low.
-l. 227-228. If you present this data, you should properly describe it in a separate paragraph. In general, I cannot understand why these data are presented. What is the point of showing cell number and optical density simultaneously?
-Figure 6. Presentation of the data on optical density is unsatisfied, because there is no information about wavelength of detection.
-Resolution of figures 5 and 6 is low.
-Figure 7: panel a is not described in the text. Describe each panel separately.
-l. 251-277. It is very difficult to follow this information. I suggest giving it in the form of a table. It does not take away that the data also should be explained in the text in a more comprehensive manner.
-There are extra figures in the subsection 3.4. Moreover, it is difficult to compare the data for different carbon sources on different graphs. I suggest omitting the figures 8-14 and presenting this data in the form of a table.
-The conclusion about the highest astaxanthin content in the algal cultures grown on the JM medium is not supported by the results. On Figure 7B the difference between astaxanthin contents in H. lacustris cultures grown on Chu-13, JM and Conway media is statistically insignificant, i.e. the same. Thus, there was no reason to select the JM medium for further experiments.
DISCUSSION
-l. 363-366. It is not supported by results (see above).
-l. 372-374. It is not a proper explanation of the H. lacustris life cycle. Green immotile vegetative cells also stop dividing. These immotile palmelloid vegetative cells turn to haematocysts (see doi.org/10.3390/md21020108).
-l. 374. What is TCA?
-l. 377. Please, explain. First, nitrates are also inorganic compounds. Second, nitrates are assimilated by the cell to produce organic compounds, not vice versa.
-l. 388-398. It is a repetition of results.
-Actually, obtained results are not discussed in the discussion section. You should explain how used organic compounds could be included in the metabolism of H. lacustris. It is especially important for cellulose, because algae do not exhibit pathways for its catabolism and it cannot be utilised as a carbon source.
Comments on the Quality of English LanguageEnglish is very weak. It should be extensively edited. There are many uncommon phrases. I provide only several serious points.
-l. 28. “”Jaworsky Medium (JM) without nitrogen produced the highest amount of astaxanthin” - culture medium cannot produce astaxanthin. Microorganisms can.
-l. 84. “most” - “the most”
-l. 86. The fact cannot be an approach. Revise.
-l. 122-128. Very poor language. Sound strange. Try to revise.
-l. 139. I would revise the language here.
-l. 223-224. Cannot understand the sentence. Revise.
-l. 243, 363. Astaxanthin production cannot be grown. Microorganisms can. Revise.
-l. 265, 269, 271. Please, understand, leucine, cellulose, glutamate and other compounds do not consist of the cells, they consist of molecules, so you cannot determine their cell numbers. What do you really mean?
-l. 278-284. Cannot understand. During the growth, the content of an organic carbon source should decrease due to its uptake by the alga. Why did you observe the increase of the organic components of the medium? I think it is an incorrect translation to English.
-l. 288-304. Please, understand, alanine, lysine and other organic compounds do not contain astaxanthin. What do you actually mean? I think it is an incorrect translation to English.
-l. 403-405. leucine, glutamate and other organic compounds cannot accumulate astaxanthin. I think it is an incorrect translation to English.
Author Response
- TITLE-Avoid shortenings in the title ( lacustris should be Haematococcus lacustris).
Answer: Thank you for the comment. The change has been made as suggested. The change is indicated by green color.
- ABSTRACT-l. 22. Write the name in full at the first mention.
Answer: Thank you for the comment. The change has been made as suggested. The change is indicated by green color.
- -l. 26. Nitrogen is a gas not fixed by chlorophytes. You should write “nitrogen source”.
Answer: Thank you for the comment. The changes have been made as suggested. The changes are indicated by green color.
- -l. 24-29. Although selection of an optimal mineral medium is an important part of each algal work, it is routine and not novel. It diverts a reader from the main goal of the work and shades the novelty.I suggest focusing on the addition of the organic sources in the abstract and reducing the information about the mineral media.
Answer: Thank you for the comment. We have corrected as the suggestion.
- INTRODUCTION-Introduction is poorly written. The information close-related to the topic is not provided, despite such works exist. Previous data on lacustris growth on different organic sources (other than acetate) must be elucidated. See e.g. works doi.org/10.3390/life14010029, doi.org/10.1016/j.algal.2019.101779, doi.org/10.1016/j.algal.2019.101497, doi.org/10.1016/j.biortech.2021.125788, doi.org/10.1016/j.aquaculture.2021.736978, doi.org/10.1016/j.biortech.2020.124421, https://doi.org/10.1016/j.biortech.2019.121334, doi.org/10.1016/j.biortech.2021.126484.
Answer: Answer: Thank you for the comment. The changes have been made as suggested. We have added new literature citing new references. The changes are indicated by green color.
- -l. 39-44. References are missing.
Answer: Thank you for the comment. We have added new literature citing new references and made the following changes: Astaxanthin, a red—orange pigment belongs to the carotenoid family. In recent years, it has acquired massive importance for its remarkable biochemical characteristics, physiological effects, and physical properties [1]. Astaxanthin is a commercially valuable compound for its application in cosmetics, food supplements, feed, and pharmaceutical field [2]. Although astaxanthin is naturally produced by a range of fungal and bacterial species, Haematococcus lacustris (Girod—Chantrans) Rostafinski (Chlorophyta) is considered the chief producer of astaxanthin (up to 5% of cell dry weight) among microalgae [3]. Large-scale production of astaxanthin from yeasts, fungi, bacteria, shrimp, special fish, etc are not worthy because of their lower astaxanthin content (less than 1% of dry weight) [4]. The changes are indicated by green color.
- -l. 47. “40-carbon” and “tetraterpene” are the same.
Answer: Thank you for the comment. We have removed “40-carbon”.
- -l. 48. “linked with isoprene units” - cannot understand this. Astaxanthin is not linked with isoprene units. It consists of eight isoprene units as a structural basis.
Answer: Thank you for the comment. We have corrected this adding a new reference. The change is indicated by green color.
- -l. 48. “β rings” - should be “β-ionone rings” (see doi.org/10.3390/md21020108).
Answer: Thank you for the comment. It has been corrected as suggested. The changes are indicated by green color.
- -l. 49. It is wrong. Astaxanthin has 13 double bonds in its π-conjugated system.
Answer: Thank you for the comment. We have corrected this adding a new reference. The change is indicated by green color.
- -l. 50-53. References are required.
Answer: Thank you for the comment. It has been corrected as suggested. The changes are indicated by green color.
- -l. 56-63. The explanation of the Haematococcus life cycle should be revised. Motile cells are spores of asexual reproduction (zoospores). Spherical palmelloid vegetative cells can divide and turn to red aplanospores independently on the zoospore formation. You should also mention under which conditions astaxanthin accumulation is observed.
Answer: Thank you for the comment. It has been corrected as suggested. The changes are indicated by green color.
- -l. 64-76. This cannot be the aim of a novel scientific study, since it has been done many times previously. This text is more suitable for methods.
Answer: Thank you for the comment. It has been corrected as suggested. The changes are indicated by green color.
- -l. 82-83. “Four of the selected organic compounds: lysine, leucine, alanine, and glutamine are amino acids” - obvious textbook knowledge! remove.
Answer: Thank you for the comment. This sentence “Four of the selected organic compounds: lysine, leucine, alanine, and glutamine are amino acids.” has been removed.
- -l. 92-94. Cannot understand the idea of this statement. What do you mean? In all living things growth is influenced by carbon metabolism. It is obvious.
Answer: Thank you for the comment. It has been corrected as suggested. The changes are indicated by green color.
- -Correct name of Spirulina maxima is Limnospira maxima, and Chlorella pyrenoidosa is actually Auxenochlorella pyrenoidosa.
Answer: Thank you for the comment. It has been corrected as suggested. The changes are indicated by green color.
- -l. 64-88. The goal of the study should be described at the end of the introduction.
Answer: Answer: Thank you for the comment. It has been corrected as suggested.
- -l. 129. It is obvious. sounds strange.
Answer: Thank you for the comment. It has been corrected as suggested. The changes are indicated by green color.
- -l. 132. “The pH was adjusted to 7” - why? For example, in the original BG-11 protocol pH is 7.5.
Answer: Thank you for the comment. We have corrected. We have mentioned the pH of all the media. Although, the cited literature for BG-11 shows a pH of 8.5 but the pH of BG-11 was adjusted to 7.5 according to new literature which we have added.
- -l. 133. “10 ml of cells” - what does it mean? How many cells?
Answer: Thank you for the comment. It is “10 ml of seed culture”. It has been corrected. The changes are indicated by green color.
- -l. 136, 146. How was it measured? At which wavelength, how long was the optical path length? Which model of spectrophotometer was used?
Answer: Thank you for the comment. It has been corrected as suggested. The changes are indicated by green color.
- -l. 139. Lux are not the units of photon flux density, they are the units of illumination. Photon flux density is 40 µmol m-2 s−1. Furthermore, the common spelling of lux is lx. You can write “illumination of 3,000 lx, which was equivalent to the photon flux density of 40 µmol m-2 s−1.”
Answer: Thank you for the comment. The change has been made as suggested. The change is indicated by green color.
- -l. 149-153. Why are only these compounds listed here? There are also a set of mineral salts and vitamins.
Answer: Answer: Thank you for the comment. It has been corrected as suggested. The changes are indicated by green color.
- -It is interesting for me, how you added cellulose to your media. It is a water-insoluble compound, so cannot be added to a homogeneous liquid system.
Answer: Thank you for the comment. We knew that cellulose is a water-insoluble compound. Although, we do not exactly know the process of cellulose solubility but cellulose dissolves in strong alkaline and acidic conditions. Additionally, cellulose dissolves by mixing with NaOH within a particular pH in lower temperature, etc. (https://doi.org/10.1007/s10570-017-1272-3)
(https://doi.org/10.5935/0103-5053.20130038)
Moreover, from the observation of H. lacustris culture with the supplementation of cellulose, we guess that cellulose dissolved partially or completely either due to the pH conditions change of the culture medium once the Haematococcus cells started to grow, divide, and metabolize; or due to the chemical ingredients present in the Jaworski medium.
- -l. 155. What is cell density? What is the difference between cell density and cell number?
Answer: Thank you for the comment. Cell density is the cell number per unit volume. In this study, we have shown results in terms of cell number.
- -Table 1. Why did you add water-free Fe-citrate in one case and the pentahydrate in another?
Answer: Thank you for the comment. We have corrected this. We added Fe-citrate pentahydrate in both. The change is indicated by green color.
- -Table 1. CO should be Co.
Answer: Thank you for the comment. The change has been made as suggested. The change is indicated by green color.
- -Table 1. Thiamin HCl should be Thiamine-HCl.
Answer: Thank you for the comment. The change has been made as suggested. The change is indicated by green color.
- -Table 1. What is the difference between B12 and cyanocobalamin? Cyanocobalamin is a form of B12.
Answer: Thank you for the comment. We have removed B12 and adjusted in cyanocobalamin row. The changes are indicated by green color.
- -Table 1. What is the difference between Ferric ammonium citrate and FeC6H5O7NH4OH?
Answer: Thank you for the comment. These are same. We have corrected this. The changes are indicated by green color.
- RESULTS-l. 178, 184. “ lacustris” should be italicised.
Answer: Thank you for the comment. The changes have been made as suggested. The changes are indicated by green color.
- -l. 187-189. If you present this data, it should be properly described. In a separate paragraph panels on the figures should be explained. Please, specify which morphological changes you observe. I do not see any obvious changes on the figures.
Answer: Thank you for the comment. This line has been removed. We have removed fig 2-4.
- -Figure 2-4: resolution of micrographs is low.
Answer: Thank you for the comment. We have removed Fig 2-4 as also suggested by another reviewer.
- -l. 227-228. If you present this data, you should properly describe it in a separate paragraph. In general, I cannot understand why these data are presented. What is the point of showing cell number and optical density simultaneously?
Answer: Thank you for the comment. We have removed l-227-228 and fig 6.
- -Figure 6. Presentation of the data on optical density is unsatisfied, because there is no information about wavelength of detection.
Answer: Thank you for the comment. We have removed fig 6.
- -Resolution of figures 5 and 6 is low.
Answer: Thank you for the comment. We have removed fig 6 and upgraded the resolution of fig 5.
- -Figure 7: panel a is not described in the text. Describe each panel separately.
Answer: Thank you for the comment. It has been corrected as suggested. The changes are indicated by green color.
- -l. 251-277. It is very difficult to follow this information. I suggest giving it in the form of a table. It does not take away that the data also should be explained in the text in a more comprehensive manner.
Answer: Thank you for the comment. It has been corrected. A new table 2 is presented for this data.
- -There are extra figures in the subsection 3.4. Moreover, it is difficult to compare the data for different carbon sources on different graphs. I suggest omitting the figures 8-14 and presenting this data in the form of a table.
Answer: Thank you for the comment. Fig 8-14 have been changed to Table 2 and Table 3.
- -The conclusion about the highest astaxanthin content in the algal cultures grown on the JM medium is not supported by the results. On Figure 7B the difference between astaxanthin contents in lacustris cultures grown on Chu-13, JM and Conway media is statistically insignificant, i.e. the same. Thus, there was no reason to select the JM medium for further experiments.
Answer: Thank you for the comment. There was some error in the error bars letters of fig 7B previously. We have corrected it.
- DISCUSSION-l. 363-366. It is not supported by results (see above).
Answer: Thank you for the comment. It is correct. Under Nitrogen depletion condition, H. lacustris grown in JM (-N) media produced astaxanthin higher than other media. Regarding error bars of the previous figure, we have corrected it and inserted a new figure.
- -l. 372-374. It is not a proper explanation of the H. lacustris life cycle. Green immotile vegetative cells also stop dividing. These immotile palmelloid vegetative cells turn to haematocysts (see doi.org/10.3390/md21020108).
Answer: Answer: Thank you for the comment. It has been revised as suggested. The changes are indicated by green color.
- -l. 374. What is TCA?
Answer: Thank you for the comment. TCA is Tricarboxylic acid cycle. It has been corrected as suggested. The changes are indicated by green color.
- -l. 377. Please, explain. First, nitrates are also inorganic compounds. Second, nitrates are assimilated by the cell to produce organic compounds, not vice versa.
Answer: Thank you for the comment. We have changed this sentence. The changes are indicated by green color.
- -l. 388-398. It is a repetition of results.
Answer: Thank you for the comment. We have rearranged. The changes are indicated by green color.
- -Actually, obtained results are not discussed in the discussion section. You should explain how used organic compounds could be included in the metabolism of lacustris. It is especially important for cellulose, because algae do not exhibit pathways for its catabolism and it cannot be utilised as a carbon source.
Answer: Answer: Thank you for the comment. We have added additional literature on the conclusion section. The changes are indicated by green color.
- Comments on the Quality of English Language-l. 28. “”Jaworsky Medium (JM) without nitrogen produced the highest amount of astaxanthin” - culture medium cannot produce astaxanthin. Microorganisms can.
Answer: Thank you for the comment. It has been revised as suggested. The changes are indicated by green color.
- -l. 84. “most” - “the most”
Answer: Thank you for the comment. The change has been made as suggested. The change is indicated by green color.
- -l. 86. The fact cannot be an approach. Revise.
Answer: Thank you for the comment. We have removed the sentence: This fact is a novel approach in this study.
- -l. 122-128. Very poor language. Sound strange. Try to revise.
Answer: Thank you for the comment. It has been revised as suggested. The changes are indicated by green color.
- -l. 139. I would revise the language here.
Answer: Thank you for the comment. It has been revised as suggested. The changes are indicated by green color.
- -l. 223-224. Cannot understand the sentence. Revise.
Answer: Thank you for the comment. It has been revised as suggested. The changes are indicated by green color.
- -l. 243, 363. Astaxanthin production cannot be grown. Microorganisms can. Revise.
Answer: Thank you for the comment. It has been revised as suggested. The changes are indicated by green color.
- -l. 265, 269, 271. Please, understand, leucine, cellulose, glutamate and other compounds do not consist of the cells, they consist of molecules, so you cannot determine their cell numbers. What do you really mean?
Answer: Thank you for the comment. It has been revised as suggested. The changes are indicated by green color.
- -l. 278-284. Cannot understand. During the growth, the content of an organic carbon source should decrease due to its uptake by the alga. Why did you observe the increase of the organic components of the medium? I think it is an incorrect translation to English.
Answer: Thank you for the comment. It has been revised as suggested. The changes are indicated by green color.
- -l. 288-304. Please, understand, alanine, lysine and other organic compounds do not contain astaxanthin. What do you actually mean? I think it is an incorrect translation to English.
Answer: Thank you for the comment. It has been revised as suggested. The changes are indicated by green color.
- -l. 403-405. leucine, glutamate and other organic compounds cannot accumulate astaxanthin. I think it is an incorrect translation to English.
Answer: Thank you for the comment. It has been revised as suggested. The changes are indicated by green color.
Overall, about 95 % of this manuscript was revised by the English language experts in the field. We have English correction certificate from https://www.textcheck.com.
Round 2
Reviewer 3 Report
Comments and Suggestions for Authors
Although the authors improved the text, a cople of important issues remain unresolved.
1. A very important conceptual question is related to the possibility of using cellulose as a carbon source for algal growth. Since it is postulated that plants cannot degrade and utilize this inert carbohydrate (see e.g., doi.org/10.1111/j.1574-6976.1994.tb00033.x), solid evidence and/or proper discussion should be provided for such a bold statement. Furthermore, the work by Hadley [25] cited by you is not about the utilization of cellulose by plants, but about its degradation by plant-associated fungi. If the results of the work were not a mistake or an experimental artifact, it could be the result of degradation by associated bacteria which are usually present in the cultures of H. lacustris, such as Flavobacterium (see e.g., doi.org/10.3390/biology10020115, doi.org/10.1016/j.micres.2022.127097).
2. Presentation of data in the form of tables (2 and 3) is more comprehensive; however, it is necessary to add the results of statistical analysis. Please indicate the data from the same statistical groups by corresponding letters as uppercase indices
Minor points
-l. 43-44. If you desctibe it here, it is also important to mention other algal sources of astaxanthin with corresponding reference(s), e.g. Bracteacoccus or Halochlorella.
-l. 58. Initially, they are green palmelloid or biflagellate.
-l. 115. The correct references to BG-11 are Stanier et al. (1971) Bacteriol. Rev., 35(2), 171-205 and Rippka et al. (1979) Microbiol. 111(1), 1-61.
-l. 133. 3000 lx is not a photon flux density. It is an irradiation. Photon flux density is 40 µmol m–2 s –1.
-. 139-141. If you do not show this data you should not describe this measurement procedure.
-Table 1. Tiamin-HCl should be Thiamine—HCl.
-l. 277. Why chlorophyll b only?
Comments on the Quality of English Language
Although English has been improved, the language is still poor. It should be extensively edited at the stage of correction.
-l. 202. The highest and the lowest.
-l. 278. accumulated?
Author Response
Main Points
A very important conceptual question is related to the possibility of using cellulose as a carbon source for algal growth. Since it is postulated that plants cannot degrade and utilize this inert carbohydrate (see e.g., doi.org/10.1111/j.1574-6976.1994.tb00033.x), solid evidence and/or proper discussion should be provided for such a bold statement. Furthermore, the work by Hadley [25] cited by you is not about the utilization of cellulose by plants, but about its degradation by plant-associated fungi. If the results of the work were not a mistake or an experimental artifact, it could be the result of degradation by associated bacteria which are usually present in the cultures of H. lacustris, such as Flavobacterium (see e.g., doi.org/10.3390/biology10020115, doi.org/10.1016/j.micres.2022.127097).
Answer: Thank you for the comment. It has been corrected as suggested. We have added supporting sentences in introduction with new references. The changes are indicated by green colour.
Presentation of data in the form of tables (2 and 3) is more comprehensive; however, it is necessary to add the results of statistical analysis. Please indicate the data from the same statistical groups by corresponding letters as uppercase indices.
Answer: Thank you for the comment. It has been corrected as suggested. The changes are indicated by green colour.
Minor points
-l. 43-44. If you desctibe it here, it is also important to mention other algal sources of astaxanthin with corresponding reference(s), e.g. Bracteacoccus or Halochlorella.
Answer: Thank you for the comment. We have corrected it as suggested. The changes are indicated by green colour.
-l. 58. Initially, they are green palmelloid or biflagellate.
Answer: Thank you for the comment. We have corrected it. The changes are indicated by green colour.
-l. 115. The correct references to BG-11 are Stanier et al. (1971) Bacteriol. Rev., 35(2), 171-205 and Rippka et al. (1979) Microbiol. 111(1), 1-61.
Answer: Thank you for the comment. We have added these two references. The changes are indicated by green colour.
-l. 133. 3000 lx is not a photon flux density. It is an irradiation. Photon flux density is 40 µmol m–2 s –1.
Answer: Thank you for the comment. We have removed 3000 lx.
-. 139-141. If you do not show this data you should not describe this measurement procedure.
Answer: Thank you for the comment. We have removed line numbers 139-141.
-Table 1. Tiamin-HCl should be Thiamine—HCl.
Answer: Thank you for the comment. We have corrected it. The change is indicated by green colour.
-l. 277. Why chlorophyll b only?
Answer: Thank you for the comment. We have changed that sentence citing new reference. The changes are indicated by green colour.
Comments on the Quality of English Language
-l. 202. The highest and the lowest.
Answer: Thank you for the comment. We have corrected it. The changes are indicated by green colour.
-l. 278. accumulated?
Answer: Thank you for the comment. We have corrected it. The change is indicated by green colour.